# A phenomenological exploration of mental toughness in decision-making: Perceptions from professional Ghanaian footballers

**Benjamin Asamoah**[1,2]*, **Adwoa Bemah Boamah Mensah**[3], **Heinrich Wilhelm Grobbelaar**[1]

1 Division of Sport Science, Department of Exercise, Sport and Lifestyle Medicine, Faculty of Medicine and Health Sciences, Stellenbosch University, Stellenbosch, Republic of South Africa, 2 Department of Nursing, School of Nursing and Midwifery, Kwame Nkrumah University of Science and Technology, Kumasi, Ghana, 3 School of Nursing and Midwifery, College of Health Sciences, Kwame Nkrumah University of Science and Technology, Kumasi, Ghana

* basamoah.chs@knust.edu.gh

## Abstract

Mental toughness is an indispensable psychological resource facilitating performance excellence but its role in decision-making within African football remains underexplored. The current study explored and described professional Ghanaian footballers perceptions on the role of mental toughness in their decision-making. A descriptive phenomenological study design was adopted. A purposive sampling technique was used to recruit 12 male Ghana Premier League players who participated in semi-structured in-depth interviews. Transcribed interviews were analysed using Colaizzi's framework to capture the essence of the participants' lived experiences. Participants described mental toughness as a dynamic context-dependent resource that influence decision-making across five domains: strategic career development, performance-based, professional development, leadership and team oriented, and general life management. Mental toughness facilitated balancing professional obligations with cultural expectations. The study provides the first phenomenological account on how mental toughness shapes decision-making in Ghanaian football. It extends mental toughness conceptualisation beyond performance enhancement to life management and culturally embedded processes shaped by structural and social realities. These insights offer implications for talent development programmes and contextually informed psychological support interventions.

## Introduction

The pursuit of excellence in sport has bolstered researchers and practitioners to explore psychological constructs that facilitate and sustain optimal performance outcomes. Among these constructs, mental toughness has attracted significant scholarly interest over the past two decades with efforts directed towards its conceptualisation,

**Data availability statement:** The raw interview data generated in this study cannot be publicly shared due to ethical restrictions protecting participant confidentiality. Participants were elite professional footballers in Ghana whose detailed accounts contain sensitive information about career decisions and personal circumstances that could enable identification despite de-identification efforts. These restrictions were stipulated in ethical approval from Stellenbosch University Research Ethics Committee for Social, Behavioural, and Educational Research (REC: SBE – 22265) and Kwame Nkrumah University of Science and Technology Committee on Human Research, Publications and Ethics (CHRPE/AP/304/21) and participant informed consent agreements. However, anonymised data underlying the specific findings of this study can be made available to qualified researchers upon reasonable request and subject to ethical review. Data access requests should be directed to the institutional representative: Dr. Ernest Adankwah, Department of Medical Laboratory Technology, Kwame Nkrumah University of Science and Technology, Kumasi, Ghana (email: eadankwah.chs@knust. edu.gh). Requesters will be required to provide institutional credentials, research justification, and data use agreements ensuring ethical data handling.

**Funding:** The author(s) received no specific funding for this work.

**Competing interests:** The authors have declared that no competing interests exist.

**Abbreviations:** GPL, Ghana Premier League; CHAN, African Nations Championship; CAF, Confederation of African Football.

operationalisation, and developmental pathways [1,2]. Mental toughness is associated with several adaptive outcomes including, effective coping, enhanced concentration, self-belief, and the ability to maintain performance under varying degrees of pressure during goal-directed pursuits [2–5]. While the growing body of positive attributions of mental toughness highlight its relevance to athletic performance, initial conceptualisations have been inconsistent and criticised for failing to distinction between what mental toughness is and what it is not [6,7]. The construct, therefore, remains conceptually contested, and its foundational components (e.g., definition, measurement, development) continue to be refined across different sport and settings [8]. With the stringent scholarly attention that mental toughness has received recently, the preponderance of evidence suggests mental toughness as a multifaceted psychological resource encompassing a blend of developed and innate values attitudes, emotions, and behaviours that enables individuals to maintain goal-directed performance in varying degrees of stress, adversity, and success [3,7,9,10].

Mental toughness has emerged as a critical psychological construct particularly in stressful situations that demand effective decision-making [9–11]. Studies intimate that mentally tough athletes display qualities that bolster effective decision-making. For instance, Crust et al.'s [12] study on high-altitude mountaineers identified that mental toughness enabled them to make rational decisions while being flexible and vigilant to the evolving circumstances within the environment. The authors further surmised that, these mountaineers demonstrated realistic thinking, analytical skills and awareness of personal limits to make sound decisions within fluctuating and high-risk circumstances. Findings from other studies indicated that mentally tough athletes are able to navigate difficult choices, including decisions that require subordinating personal interests for safety or team-related considerations [3,7,13]. These findings suggest a dynamic interplay between mental toughness and decision-making, where mental toughness fosters a balanced approach that integrates perseverance with rational evaluation of situational demands [13]. The overreliance on mental toughness may, however, inadvertently engender rigid persistence, leading to poor decisions and undesirable outcomes [12].

Within the African context, football carries deep cultural and social value, operating as a vehicle for national pride and personal advancement [14,15]. For many indigenous Ghanaian footballers their progression through the competitive league structures is marked by financial constraints, infrastructural challenges, and strong cultural expectations to succeed. These contextual experiences influence career pathways and often shape how they approach decisions about their goals and conduct, particularly when they must balance personal aspirations with team obligations and community expectations. The tension between structural and cultural dynamics are integral to athletes' lived experiences and create a unique atmosphere in which mental toughness is both tested and defined [14,15]. Yet, limited attention has been given to how these contextual realities influence the meaning and manifestation of mental toughness in Ghanaian football. This is critical as Gucciardi et al. [16] posited that, the manifestation of mental toughness may be contextually driven and influenced by the specific demands and cultural values of the sporting environment. Moreover, Fawcett

[17] and Blodgett *et al*. [18] argued that psychological constructs must be grounded in the sociocultural realities of those who experience them, as individuals differ in how they interpret, navigate, and respond to situational challenges. This experiential heterogeneity is underpinned by the social and cultural resources athletes draw upon to construct meaning of their sporting realities rather than relying solely on theoretical consensus [8,19]. This underscores the need to explore the concept within diverse cultural lenses to deepen understanding of its variability and relevance.

Stemming from the aforementioned considerations, it is surprising that no study has systematically explored how mental toughness shapes decision-making among African footballers who have to navigate unique sociocultural and economic pressures. This gap is particularly salient given that decision-making extends beyond in-game tactics to encompass professional development and personal welfare. The present study, therefore, explored and described Ghanaian footballers' perceptions on how mental toughness influence decision-making in their sporting and personal lives. This study aims to provide a contextually grounded understanding of the mental toughness construct within Ghanaian football culture, addressing a significant gap in empirical research on how mental toughness is interpreted and manifested in this underexplored context.

## Materials and methods

### Study design and philosophical orientation

A descriptive phenomenological study design was used to explore professional Ghanaian footballers lived experiences and perceptions of the role of mental toughness in their decision-making across their professional and personal lives. Phenomenology, as both a philosophical orientation and a research method is geared towards the description of a phenomena as consciously experienced without theoretical imposition [19,20]. Central to this approach is Husserl's transcendental orientation, emphasising epoché and eidetic reduction, which guide the researcher in returning "to the things themselves" to uncover the essential structures of participants' lived experience [21,22]. This line of enquiry requires researchers to suspend their natural attitudes, theoretical assumptions, and prior knowledge to allow the phenomenon to reveal itself in a pure form through the participants' descriptions of their lived experiences [23]. Descriptive phenomenology was intentionally selected as it provides a rigorous and systematic framework for exploring depth, nuance, and contextual richness of lived experiences, particularly in areas where limited prior research exists [24].

### Study setting

The study was conducted across four regional capitals in Ghana – Accra (Greater Accra Region), Kumasi (Ashanti Region), Sunyani (Bono Region), and Cape Coast (Central Region), which collectively housed most of the participating teams during the 2020/2021 Ghana Premier League (GPL) season. The GPL was intentionally selected as the primary recruitment platform given its status as Ghana's preeminent football competition featuring the best players, coaches and technical staff. The decision to use footballers competing in the GPL is consistent with the prevailing notion that, the facets of mental toughness are better manifested among athletes who compete at the elite level [10,25]. Also, the selection of the GPL as the recruitment context is consistent with Patton's [26] recommendation to seek rich and detailed information from participants whose lived experiences produces contextually grounded insights into the phenomenon under investigation.

### Population, sampling and sample size

The study employed a criterion-based purposive sampling approach to enrol indigenous Ghanaian footballers with demonstrable mental toughness within the GPL. Participant enrolments were conducted in collaboration with coaches (recruitment links) from the four study sites, selected for their positional authority and extensive league experience. These coaches played a dual methodological function: (a) as cultural interpreters who could articulate context-specific understanding of mental toughness, and (b) as gatekeepers with insider knowledge required to identify players whose

lived experiences aligned with the study focus. This approach is consistent with established methodological practices in mental toughness research (e.g., [3]), that acknowledges the context-embedded nature of the construct and the value of expert informants in identifying suitable participants [2]. To minimise potential selection bias associated with gatekeeper-mediated recruitment, we adopted a cross-validation procedure where players whose names consistently appeared across the coaches' nominated lists were invited to participate in the study. This triangulation strategy reduced individual coach bias and ensured that the final recruitment pool reflected a shared understanding of mentally tough players within the Ghanaian football ecosystem. We acknowledge that coach-mediated recruitment may still introduce some bias; however, this approach is deemed methodologically appropriate for phenomenological research, which prioritises depth and contextual richness over representativeness.

The inclusion criteria required participants to: 1) be active or retired indigenous Ghanaian footballers who had played club football exclusively in Ghana; 2) if active, be officially registered with a GPL club; 3) if retired, have ended their professional career within the past 15 years to ensure adequate recall; and 4) have a minimum of five years of GPL playing experience to ensure adequate competitive exposure to high-level competitive decision-making beyond the initial adaptation phase. After coaches consultations, 50 cross-validated nominees were approached by the lead author. Twenty players expressed interest to take part in the study, and by the 10th interview no new substantive insights emerged pertaining to the breath or specificity of themes indicating we achieved data saturation [27]. We conducted two additional interviews to confirm this determination, resulting in a final sample of 12 participants. The final cohort comprised 12 male players occupying different positional roles: one goalkeeper, three defenders, six midfielders, and two forwards. Eight of the players were active registered GPL players and four were retired professionals. This variability in positional roles and career stage enhanced the depth of experiential accounts, as the cognitive, emotional, and tactical demands associated with mental toughness differ across playing positions and competitive phases. Collectively, this diversity allowed both contemporary and retrospective insights on mental toughness in decision-making to be captured.

## Data collection

The study used an in-depth semi-structured interview guide developed in alignment with the research objective and based on established mental toughness literature [3,7]. The guide comprised two sections, the first section encompassed participants' demographic information and was closed-ended (e.g., age, GPL playing experience, playing status, playing position), and the second section consisted of open-ended questions designed to elicit participants lived experiences of mental toughness and decision-making. The interview style embraced the phenomenological principle of highlighting participants as co-constructors of knowledge, allowing naturalistic dialogue that gives fidelity to their experiential narratives [28].

Prior to the interviews, we pretested the interview guide with three players (excluded from the final analysis) to assess the question flow, conceptual clarity, and address practical concerns of the guiding questions accordingly. Following the pretesting, the authors engaged an expert with extensive technical experience within the Ghanaian football ecosystem to review the guide for sociocultural appropriateness and contextual relevance. This consultation resulted in the reordering of the questions to enhance cultural sensitivity and optimise participant engagement.

The main guiding questions were a) Please describe as detailed as possible what mental toughness in Ghanaian football is. Can you provide a definition, phrase, or quote to describe it? b) Please describe a critical situation where you had to make an important decision in your playing career; c) Can you describe how mental toughness influences your decision-making in your professional career and personal life? We then used specific follow-up probes to encourage deeper exploration.

All the interviews were conducted face-to-face by the first author, was audio-recorded, and supplemented with field notes documenting all nonverbal cues, and interviewer reflections. The interviews were conducted at participants' homes

at times and dates of their choosing. Interview duration lasted 64 minutes on average (range 37–120 minutes). Participant recruitment commenced on 30 October 2021 and ended on 17 June 2022.

## Ethical considerations

We obtained ethical approval from the Stellenbosch University Research Ethics Committee for Social, Behavioural, and Educational Research (REC: SBE – 22265) and the Kwame Nkrumah University of Science and Technology Committee on Human Research, Publications and Ethics (CHRPE/AP/304/21) for the study. The authors also collaborated with the technical directorate of the Ghana Football Association who granted permission to use their platform (GPL) to recruit participants for the study. During recruitment, participants received detailed information regarding the study's purpose, potential benefits and risks, and audio-recording procedures. This enabled them to make informed decisions regarding voluntary participation. Although there were no direct benefits, participants were informed that their contributions would advance emic understanding of mental toughness in the Ghanaian football and potentially inform culturally tailored intervention programmes for young players. The participants were informed of their right to voluntary participation, the prerogative to decline responding to specific questions, and the right to withdraw from the study at any point without prejudice. All participants provided written informed consent documenting their voluntary agreement to participate. Participants were assured of their anonymity and confidentiality in all future publications and reports. Alphanumeric codes (LP1 – LP12) were given to each participant to maintain anonymity. All anonymised study data will be destroyed after five years post-study completion.

## Data analysis

Data collection and analysis proceeded concurrently, consistent with the phenomenological approach's iterative nature. All interviews were audio-recorded to fully capture participants' lived experiences and facilitate comprehensive transcription. At the end of each interview, recordings were reviewed and securely saved on password-protected devices. The interviews were conducted in Twi (local vernacular) and translated into English by a professional transcriber proficient in both languages. To ensure translation accuracy and cultural nuance preservation, the second author (A.B.B.M.) reviewed a random sample of recordings and transcripts to verify fidelity between the original participants' accounts, and the English translations. Reflexivity was ensured through different strategies implemented throughout the research process. Prior to data collection and analysis, the authors engaged in regular reflexivity meetings to actively explore potential analytic challenges, identify possible biases on the mental toughness concept, and discuss interpretive assumptions that might influence participant narratives. The lead author kept reflective memos after each interview to capture emerging biases and interpretive shifts as themes began to crystallise. These reflexive practices were integrated into ongoing bracketing efforts, wherein the authors suspended presuppositions about mental toughness to allow participants' meanings to emerge authentically.

The transcribed data were then imported into NVivo 14 [29] for data management and analysis. Consistent with the study's descriptive phenomenological stance, Colaizzi's [19] seven-stage systematic and rigorous analytic framework was employed:

Step 1: The authors (B.A. and A.B.B.M.) engaged in close data immersion through multiple iterative readings of interview transcripts, and consciously suspending personal assumptions and immediate judgements to allow authentic participant voices and meaning to emerge organically.

Step 2: The authors performed separate inductive coding to identify all data segments relevant to the mental toughness phenomenon through line-by-line transcript analysis. This dual coding approach facilitated analyst triangulation and minimised individual researcher bias [30].

Step 3: The analysts meticulously extracted relevant statements and phrases, conducting in-depth contextual analysis to illuminate the underlying meanings and essence of mental toughness and decision-making experiences. This entailed scrutinising every coded segment within its broader narrative context, considering participants' linguistic choices and cultural references.

Step 4: Formulated meanings were grouped into initial themes and theme clusters guided by the conceptual relationships and shared characteristics in participants' accounts. This involved constant comparison where emerging themes were progressively refined and reorganised as additional data were analysed.

Step 5: Patterns and relationships among themes were systematically woven together to create an overarching description of the fundamental structure of the role of mental toughness in decision-making while preserving the accuracy of participants' distinctive idiosyncratic experiences.

Step 6: Detailed descriptions were condensed into brief, dense statements that captured the underlying structure of mental toughness and decision-making process. Field notes and reflexivity records were used to corroborate the analysis and interpretations.

Step 7: Member checking was undertaken by returning summarised responses to three randomly selected participants to verify the accuracy of the fundamental structure statements of their experiential accounts. This process yielded no request for modification, which supported the credibility of the findings.

### Trustworthiness and credibility

Credibility was achieved through implementing triangulation, confirmability, and transferability strategies [31–33]. Data source triangulation was achieved through the integrated perspectives from both active and retired Ghanaian footballers, whose professional records at the Ghana Football Association provided context to their playing experiences. This approach offered comprehensive temporal view of mental toughness in decision-making. Supplementary data sources including field notes documenting non-verbal cues, contextual concerns, and the interviewer's reflections, were referred to during the analysis to give multiple analytical lenses to the descriptions of the participants lived experiences. Moreover, analyst triangulation was implemented through collaborative analysis wherein the first and second authors systematically compared and discussed codes until reaching consensus. This collaborative approach minimised individual researcher bias and enhanced the interpretative validity of the findings [34]. Three randomly selected participants validated their interview transcripts one-week post-transcription. No revisions were requested by the participants, indicating that the transcripts were an accurate reflection of the interview content. For transferability, rich, exhaustive descriptions of the research context, participants' characteristics, analytical processes, and study findings were explicitly provided to enable other researchers to assess applicability to their contexts and settings.

## Results

### Participants

Twelve indigenous Ghanaian professional footballers with an average age of 33.25 years (range 22–60 years) playing across different positions (i.e., goalkeeper = 1, defenders = 3, midfielders = 6, forwards = 2) participated in the study. Eight of the participants were currently active footballers in the GPL, while four were retired at the time of the data collection. Notably, half of the participants (n = 6) had competed in the African Nations Championship (CHAN), a biennial Confederation of African Football (CAF) tournament, exclusively featuring footballers from the participating countries' domestic leagues. The participants' GPL playing experience averaged 9.67 years (range 5–18 years).

## Summary of themes

Overall, 16 lower-order themes subsumed under five overarching themes that characterise Ghanaian footballers' perception of the role of mental toughness in their decision-making emerged. The five emergent decision-making themes encompass: 1) strategic career development; 2) performance-based; 3) professional development; 4) leadership and team-oriented; and 5) general life management decision-making. The overarching themes, lower-order themes, and their descriptions are detailed in Table 1, with selected in-vivo quotes provided in the subsequent sections.

**Strategic career development decision-making.** This overarching theme describes how mental toughness emerged as an internal regulatory mechanism enabling players to chart intentional career paths by prioritising long-term strategic gains over immediate gratification. Participants described how mental toughness shielded them from impulsive decisions under external pressure and enabled them to make meticulous evaluation of opportunities that facilitated future career prospects. This regulatory function is exemplified in one player's response to an enticing transfer offer:

*I thought that was not the right time for me to leave [name of club] to join [name of club] despite pressure from my agent, family, and friends to make the switch following my win as the league's best player during that season. It was mental toughness that enabled me to stay resolute in the face of such pressure.* (Participant 1)

This capacity to maintain strategic patience merged with players' ability to sustain clarity during career transitions and adapt to evolving circumstances. Mental toughness functioned as a self-reframing resource, allowing for detailed opportunity analysis while maintaining flexibility to embrace challenges as apparent in the following quote:

*In my transfer to [name of club], I had been asked to play a position which was not my natural position… I composed myself and said to myself that, I am mentally tough and possess the adaptability to thrive in the new position.* (Participant 4)

Players further described mental toughness as a stabilising force in career navigation, allowing them to remain focus while adapting smoothly to changes. This adaptability extends beyond tactical flexibility to include career management as evident in this quote:

*Football careers are dynamic with frequent changes in coaches, teammates, and playing conditions. Mental toughness allows me to adapt to these changes smoothly and make decisions that resonate with evolving circumstances.* (Participant 12)

These accounts reveal mental toughness functioning as a temporal bridging mechanism, resourcing players to withstand immediate pressures while sustaining long-term strategic vision. This is particularly important in environments where socioeconomic constraints might otherwise drive premature career decisions.

**Performance-based decision-making.** Performance-based decision-making emerged as the most prominent theme, underscoring mental toughness central role in real-time competitive situations. The lower-order theme "focused and strategic in-game decision-making" had the highest frequency of significant participant quotes, revealing mental toughness as essential for maintaining optional decision-making capacity amid intense physical, emotional, and tactical pressures of competitive match play. This finding illuminates mental toughness functioning not merely as stress tolerance but as an active cognitive-regulatory mechanism: Participant 1 articulated this relationship as:

*When you are fatigued in the 80$^{th}$ minute and need to decide whether to make that bustling run or hold your position, it is mental toughness that helps you choose wisely. It allows you to stick to the game plan when things are not going well rather than panicking and abandoning strategy.*

**Table 1. Mental toughness and decision-making among Ghanaian footballers: Lower-order themes, theme descriptions, and overarching themes.**

| Lower-order themes | Theme descriptions | Overarching themes |
|---|---|---|
| **Patience regarding career advancement (n = 4)** | Describes how mental toughness enables athletes to prioritise long-term career benefits over immediate gains, and maintain focus despite external pressures and challenges. | Strategic career development decision-making |
| **Clarity and strategic career choices (n = 6)** | Highlights athletes' ability to remain focused and make well-informed decisions, whilst under pressure. It reflects the importance of evaluating career options thoroughly, prioritising personal development goals such as education or skill-development, alongside a football career. | |
| **Embracing career fluctuation through adaptive decision-making (n = 1)** | Describes how mental toughness enables players to view career fluctuation as opportunities for growth, facilitating adaptive decision-making in response to changing football dynamics and conditions. | |
| **Embracing positional adaptability through intensive self-development (n = 1)** | Acknowledges how mental toughness enables players to embrace and excel in new team roles through committed self-development, transforming positional challenges into growth opportunities. | |
| **Focused and strategic in-game decision-making (n = 10)** | Captures how mental toughness equip players to maintain effective decision-making during matches despite physical fatigue and emotional pressure, ensuring their actions align with the game objectives. | Performance-based decision-making |
| **Balancing risk and caution through contextual assessment (n = 3)** | Describes the player's ability to balance tactical risk-taking, where mental toughness influences rational evaluation of match situations, enabling calculated choices over impulsive actions based on contextual demands. | |
| **Embracing high-stakes action in crucial moments (n = 7)** | Characterises mental toughness as a key psychological resource that facilitates players' delivery of decisive actions during high-stake moments, enabling the selection of potentially transformative choices over conservative options despite performance pressure and potential negative outcomes. | |
| **Maintaining decisive action-taking despite setbacks (n = 7)** | Describes mental toughness as an important psychological mechanism that sustains players' capacity for decisive action despite performance setbacks, thereby preserving decision-making confidence and preventing regression into passive response patterns following adverse experiences. | |
| **Committing to rigorous preparation despite adversities (n = 3)** | Describes mental toughness as a performance enabling mechanism that preserves players' commitment to meticulous preparation standards despite environmental constraints, facilitating the pursuit of long-term development over situational comfort or convenience. | Professional development decision-making |
| **Opting for challenge-driven growth over comfort (n = 2)** | Underlines mental toughness as a growth enabling factor that propels players toward deliberate challenge engagement, assist the reframing of demanding situations as developmental opportunities rather than threats to established performance patterns. | |
| **Prioritising self-improvement over immediate gratification (n = 6)** | Illuminates mental toughness as a self-regulatory mechanism that facilitates player's adherence to development choices, enabling their prioritisation of long-term performance enhancement over immediate gratification through sustained discipline and targeted skill acquisition. | |
| **Cultivating disciplined decision-making across all spheres (n = 2)** | Describes how mental toughness influences goal-directed behaviour across professional domains, facilitating rational decision-making process over emotional reactivity while aligning daily choices with career objectives. | |
| **Asserting constructive dissent despite hierarchical pressure (n = 2)** | Describes how mental toughness manifests as a status bridging process allowing players to articulate constructive dissent within hierarchical team structures in advancing collective improvement over personal preservation despite inherent social risks. | Leadership and team-oriented decision-making |
| **Exercising situational leadership discernment (n = 3)** | Portrays how mental toughness functions as an adaptive leadership process that enables context sensitive behavioural modulation where players seamlessly shift between assertive and supportive inclinations in response to team dynamics and individual developmental needs. | |
| **Balancing life and career through deliberate choices (n = 1)** | Stresses the functioning of mental toughness as a life domain balancing mechanism that enables deliberate boundary management and resource allocation to enhance well-being amid competing personal and professional demands. | General life management decision-making |
| **Prioritising career sustainability over family expectations (n = 1)** | Showcases the mediating role of mental toughness as a socio-cultural boundary regulation mechanism that enables players to navigate familial financial obligations, facilitating sustainable career decisions despite cultural pressures and extended family expectations. | |

Note: n represents the number of participants that cited a particular lower-order theme.

Mental toughness further enabled players to balance calculated risk-taking with caution during high-stakes situations, facilitating active emotional redirection rather than impulsive responsiveness. One player's reflection of a provocative match situation illustrates this regulatory capacity:

*I remember this match against [name of club]. One of their players was trying to provoke me, stepping on my toes off the ball, whispering insults. A less mentally tough player might have retaliated and gotten a red card. But I channelled that frustration into my performance instead. Ended up assisting the winning goal. That is the best response.* (Participant 11)

These experiences reveal mental toughness as enabling contextually appropriate risk assessment while maintaining decisive action despite frustration, illustrating restraint without passivity during high pressure match situations. This observation suggests mental toughness operates as a decision-making facilitator that integrates emotional regulation with strategic cognition, extending the rigid perseverance highlighted in Western conceptualisations.

**Professional development decision-making.** Mental toughness influenced players' commitment to continuous improvement through interconnected decisions about training, preparation, and growth. Players described mental toughness propelled them toward challenging situations, enabling them to reframe situations as developmental opportunities rather than threats. The respondents actively sought and created challenging training environments to identify improvement areas, relegating short-term gratification to uphold rigorous standards across professional development spheres.

*When everyone is cutting corners during fitness drills, mental toughness helps you to decide to push through properly. When you are tempted to skip extra practice because you are tired, it helps you decide to put in those extra hours.* (Participant 11)

Participants perceived mental toughness as enabling integrated decision-making that aligned their professional activities with long-term development goals, facilitating disciplined choices across multiple domains as evident in the expression below:

*Whether it is how I approach training, handle criticism, or manage off-field distractions, it all comes down to staying disciplined and not letting emotions cloud my judgement.* (Participant 4)

This theme highlights mental toughness as a self-regulatory mechanism that sustains commitment to developmental processes despite immediate discomfort or situational adversity. This function is particularly significant in resource-constrained contexts where training facilities and support are limited.

**Leadership and team-oriented decision-making.** This theme explains how mental toughness guided team-centred decisions-making, revealing an often-overlooked interpersonal dimension of the construct. Players described mental toughness equipping them to deal with complex interpersonal situations, including providing constructive critical feedback despite social risks within hierarchical team structures. They explained how mental toughness empowered them to assert constructive dissent when it served the collective good of the team, even when doing so created interpersonal tension or risked social cohesion. This capacity to bridge status differences and provide critical feedback across hierarchical boundaries reflects mental toughness as a resource for managing social dynamics. This is illustrated in the narratives below:

*In the dressing room, mental toughness is crucial for decision-making as a senior player. Sometimes you must decide whether to confront a teammate or support them quietly.* (Participant 11)

*In the dressing room, mentally tough players often become natural leaders. They're the ones who speak up when things are tough, offering encouragement or constructive criticism when needed. They help manage conflicts and keep the team united, especially during tough moments in the season.* (Participant 1)

Players further described mental toughness as facilitating situational leadership through behavioural modulation by alternating between assertive and supportive styles depending on specific situational and individual demands. They gauge when direct confrontation advanced team goals versus when measured behind-the-scenes support more effectively promoted individual development and team growth. This adaptive quality extends understanding of mental toughness beyond intrapersonal attributes to encompass complex interpersonal discernment and context-embedded leadership.

**General life management decision-making.** This theme portrayed how mental toughness influenced players' capacity to balance professional responsibilities with complex cultural and personal expectations, especially within the Ghanaian milieu where family and cultural obligations can be high. This theme revealed mental toughness facilitating the management of competing demands while maintaining sustainable career development. Players explained how mental toughness empowered them to set boundaries, make balanced life choices, and handle financial pressures, stressing the cultural complexity of professional football in Ghana.

*As footballers in Ghana we often become breadwinners early. Mental toughness helps you make tough decisions about money, who to help, when to say no.* (Participant 11)

Players described mental toughness assisting difficult decisions regarding resource allocation and family support obligations, enabling exercise of personal boundaries while upholding family responsibilities despite sociocultural expectations and emotional pressures. They explained how mental toughness enabled boundary management and resource allocation across competing personal obligations, balancing career sustainability with family expectations to make long-term career decisions that advances both the personal achievement and family welfare. This is shown in the quotes below:

*Dealing with family pressures is another off-field challenge. In Ghana, when you become a professional player, many family members start depending on you financially. Managing these expectations while trying to save for your own future requires strong mental resolve.* (Participant 1)

*Then there's the financial side of things. Suddenly you're making good money, everyone wants a piece of it. Family members you've never met before are calling, asking for help. It's hard to say no sometimes, but you've got to be smart, think about your future. That takes mental toughness to make those tough decisions.* (Participant 3)

This theme surmises mental toughness as functioning as a sociocultural boundary regulation mechanism – perhaps the study's most distinctive contribution by illuminating how the construct enables the navigation of cultural obligations unique to collectivist, resource-constrained contexts.

## Discussion

The study explored professional Ghanaian footballers perceptions of how mental toughness influences decision-making across their sporting and personal lives. The findings indicate that mental toughness operates as a multidimensional psychological resource that shapes decisions across five distinct domains: strategic career development, performance-based, professional development, leadership and team-oriented, and general life management decision-making. These results broaden conventional conceptualisation that accentuate mental toughness primarily as a performance-enhancing attribute [10,25]. Instead, the construct appears to operate as a comprehensive decision-making resource that influence choices across different timeframes and life contexts. To our knowledge, this study is the first systematic exploration of mental

toughness and decision-making within African football, offering culturally situated insights that corroborate and advance existing theoretical frameworks.

Mental toughness emerged as a key facilitator of strategic career decision-making, particularly when players confronted pressures that could compromise long-term career trajectories in the face of immediate gratifying enticing offers. This aligns with research intimating that psychological resources are essential in managing career transitions [35,36]. The present findings advance literature by identifying specific mechanisms through which mental toughness operates. Participants described mental toughness as an internal regulatory system that enables them resist pressures from agents, family members, and peers when such pressures conflict with long-term career goals. This regulatory function likely operates through executive control pressures that support future-oriented thinking and the capacity to delay short-term rewards [37,38]. The ability to maintain strategic focus despite significant sociocultural and economic pressure is distinctively meaningful in Ghana, where players often manage intense financial demands from extended family and converging community obligations [15,39]. These findings surmise that mental toughness may hold high relevance in collectivist contexts like Ghana, where economic constraints and interconnected social networks create competing demands on athletes' decision-making. This interpretation is consistent with Gucciardi's [2] argument that mental toughness is shaped by contextual influences. The results also reveal a key nuance, that is, while the underlying mechanism for self-regulation and delayed gratification may be universal, the situations the necessitates its manifestations are culturally specific.

Performance-based decision-making surfaced as the most commonly referenced domain. This reinforces established evidence that mental toughness enhances performance under pressure [4,7,40]. The participants' account of maintaining strategic focus despite fatigue or provocation support Crust and Clough's [40] description of mental toughness as a buffer against suboptimal performance under stressful conditions. The present findings further develop this understanding by identifying the cognitive and emotional processes that underpin this buffering effect. The respondents underscored mental toughness as facilitating emotional regulation by reframing frustration into productive effort rather than reactive behaviour. They also emphasised its role in attentional control by maintaining focus on task relevant demands while suppressing distracting provocations. This dual-process mechanism reveal that mental toughness promotes effective performance not simply through persistence, but through the integration of regulatory capacities [41,42].

The subtheme of "embracing high-stakes action in crucial moments" is particularly revealing, in that it advances mental toughness as not only enabling players to sustain performance under pressure but also facilitate risk-taking when appropriate. This finding advances the idea promulgated by Crust et al. [12] that an overreliance of mental toughness may encourage rigid perseverance that leads to poor decisions and negative outcomes. The current results demonstrate that when mental toughness is shaped by contextual sensitivity and situational appraisal, it supports adaptive responses rather than inflexible decision-making occasioned by the maladaptive rigidity described by Crust et al. This adaptive expression may be guided by the realities of Ghanaian football, where limited resources and narrow margins for error require players to develop sophisticated risk assessment capacities. Alternatively, it may also represent a universal but underexplored facet of mental toughness that warrants inclusion in theoretical models. The capacity to think clearly and sacrifice short-term rewards for sustained career development exhibits how mental toughness might be uniquely valuable in Ghana, where players contend with strong community expectations and belief that football provides a pathway to economic advancement [15].

The findings related to professional development corresponds with Duckworth's [43] conceptualisation of grit, indicating conceptual overlap theoretically with perseverance toward long-term goals. Our findings also extend Toering and Jordet's [44] work on self-regulation by illustrating how mental toughness facilitates commitment to development process despite discomfort, environmental constraints or limited resources. The inclination of mentally tough players to prioritise self-development at the expense of immediate gratification demonstrate the role of mental toughness as a regulatory resource for delayed gratification in pursuit of skills and long-term growth. This predisposition appears relevant across contexts, though its salience may be heightened in resource-constrained environments where athletes must overcome significant obstacles such as inadequate facilities, limited coaching support and competing demands on their time.

Leadership and team-oriented decision-making emerged as a distinct conceptual domain, marking a meaningful extension of current mental toughness conceptualisation, which often emphasises individual rather than interpersonal attributes [45]. These findings concur with Gucciardi *et al.*'s [46] proposition that mental toughness involves both interpersonal and intrapersonal elements. The present study provides empirical examples of how these interpersonal capacities appear in decision-making situations. Participants highlighted mental toughness as instrumental in navigation of hierarchical team structures, enabling constructive dissent when team performance or cohesion required it, and facilitating adaptive shifts between assertive and supportive leadership behaviours. This interpersonal dimension may reflect cultural expectation in collectivist contexts, where social harmony and respect for hierarchy carry substantial weight [47]. The ability to assert constructive dissent despite hierarchical pressure requires careful social judgment. This understanding stresses an underrepresented interpersonal dimension of mental toughness that enhances not only individual performance but also contribute to team functioning and psychological safety as advanced by Edmondson and Lei [48]. The dimension of 'interpersonal process" warrants integration into mental toughness frameworks as the finding suggest the construct facilitates not only individual dispositions but collective functioning and team development.

A key contribution of this study is the identification of general life management decision-making as a domain where mental toughness functions as a sociocultural boundary regulation mechanism enabling players manoeuvre competing professional and familial obligations. While all athletes contend with balancing work and personal life challenges, the cultural context of Ghanaian football, where players often assume breadwinner responsibilities for extended families early in their career create a unique decision-making demand [39,49]. Mental toughness enabled players to manage overlapping demands by setting clear boundaries around family financial requests, allowing them to honour cultural obligations without undermining the sustainability of their careers. This finding extends current understanding of mental toughness beyond performance contexts into broader life management shaped by cultural and economic realities. Studies on African footballers abroad has similarly revealed that players navigate deep obligations associated with kinship support, and community expectations, often described as remittances or black tax. These obligations carry both emotional meaning and material strain, influencing well-being and career trajectories [39,49,50]. However, such studies have primarily focused on the sociological patterns of giving and the associated costs and benefits rather than the psychological underpinnings that enable players to regulate these pressures. The current study advances the body of knowledge by surmising that mental toughness can function as a boundary regulation mechanism that helps indigenous Ghanaian players balance legitimate but competing personal and familial demands.

Consequently, these findings indicate that while the underlying regulatory processes occasioned through myriad demands may transcend sports and contexts, the circumstances that engender its expressions are markedly prominent in collectivist and resource-constrained environments where familial interdependence is central.

## Applied implications

The findings present important implications for talent development and applied practice in settings where athletes are exposed to significant sociocultural and economic pressures. Talent development programmes should incorporate mental toughness training specifically focused on strengthening decision-making capacities as opposed to focusing solely on performance enhancement. Designed programmes can achieve this by providing structured opportunities for athletes to practice future-oriented decision-making and delayed gratification. Additionally, workshops can be organised to address family financial expectations and culturally appropriate boundary setting. Moreover, training simulations that require context-sensitive judgement can further develop these skills by immersing athletes in realistic scenarios where they must navigate complex, real-world decisions.

Practitioners should adopt culturally informed approaches that situate mental toughness as operating within the larger sociocultural contexts in which athletes function. Assessment practices should consider decision-making across the sport and non-sport milieu and acknowledge the cultural and familial pressures as legitimate challenges necessitating mental

toughness rather than see it as distractions from sports. These implications are especially relevant for young footballers in resource-constrained environments who face overwhelming external pressures to prioritise short-term financial gains. Thus, providing athletes with the psychological resources that support long-term decision-making may help them sustain their development despite converging demands from their immediate social environment.

## Conclusion

The study advances mental toughness as a complex multidimensional psychological resource that extends beyond performance enhancement conceptualisations and highlight it as an adaptive capacity to make effective decisions across career management, performance optimisation, development progression, leadership execution, and general life management domains. The findings conveyed the importance of mental toughness as a self-regulation process for athletes to deal effectively with immediate in-game decisions and strategic long-term planning under significant cultural, socioeconomic, and professional pressures. The cultural influences within the Ghanaian football context situated mental toughness at the nexus between the demands of professional sport and the sociocultural expectations to which footballers must deftly manoeuvre to surmount the inherent challenges therein. The findings constitute a vital step towards a comprehensive understanding of the mental toughness phenomenon and offer relevant implications for coaches and sports practitioners in a culturally diverse and high-pressured sporting environment.

## Limitations and future directions

The study offers unique insights into Ghanaian footballers' perceptions of mental toughness in decision-making. As with any study, there were limitations that should be acknowledged. For instance, the study's phenomenological stance captured lived experiences but did not quantify mental toughness or establish causal relationships. Future studies should employ mixed methods approaches in assessing whether higher levels of measured mental toughness have any influence on specific decision-making patterns or outcomes. Similarly, while the focus on mental toughness in decision-making among Ghanaian premier league athletes adds meaningful insights to existing knowledge; future studies must explore these constructs across different competitive tiers and conduct cross-cultural comparison between African and Western contexts. Such investigations would capture a broader spectrum of mental toughness in Ghanaian football while distinguishing between which elements of mental toughness and decision-making are universal (etic) and which elements are culturally determined (emic). Future research should explore how mental toughness development targeting decision-making can be integrated into talent development programmes. Longitudinal studies should examine how such interventions shape career progression and how mental toughness influence adaptation to evolving competitive demands as their careers progress.

## Supporting information

**S1 Questionnaire. Inclusivity in global research questionnaire.**
(DOCX)

## Acknowledgments

The authors would like to express their gratitude to all the footballers who shared their valuable experiences. We also thank the Ghana Football Association for their support in facilitating access to the participants.

## Author contributions

**Conceptualization:** Benjamin Asamoah, Heinrich Wilhelm Grobbelaar.
**Data curation:** Benjamin Asamoah.

**Formal analysis:** Benjamin Asamoah, Adwoa Bemah Boamah Mensah, Heinrich Wilhelm Grobbelaar.

**Investigation:** Benjamin Asamoah.

**Methodology:** Benjamin Asamoah, Adwoa Bemah Boamah Mensah, Heinrich Wilhelm Grobbelaar.

**Project administration:** Benjamin Asamoah.

**Resources:** Benjamin Asamoah.

**Software:** Benjamin Asamoah.

**Supervision:** Adwoa Bemah Boamah Mensah, Heinrich Wilhelm Grobbelaar.

**Visualization:** Benjamin Asamoah.

**Writing – review & editing:** Benjamin Asamoah, Adwoa Bemah Boamah Mensah, Heinrich Wilhelm Grobbelaar.

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
