## [Decision Letter · Decision Letter 0]

27 Oct 2025

Dear Dr. Asamoah,

Thank you for submitting your manuscript to PLOS ONE. After careful consideration, we feel that it has merit but does not fully meet PLOS ONE’s publication criteria as it currently stands. Therefore, we invite you to submit a revised version of the manuscript that addresses the points raised during the review process.

Academic Editor Comments: Both reviewers acknowledged the study’s potential and overall soundness. Reviewer #1 suggested minor improvements to clarity and structure, while Reviewer #2 recommended major revisions to strengthen the theoretical framework, methodological transparency, and logical coherence. A major revision is therefore requested before further consideration.

We look forward to receiving your revised manuscript.

Kind regards,

Wanli Zang

Guest Editor

PLOS ONE

Journal Requirements:

4. In this instance it seems there may be acceptable restrictions in place that prevent the public sharing of your minimal data. However, in line with our goal of ensuring long-term data availability to all interested researchers, PLOS’ Data Policy states that authors cannot be the sole named individuals responsible for ensuring data access (http://journals.plos.org/plosone/s/data-availability#loc-acceptable-data-sharing-methods).

Reviewers' comments:

Reviewer's Responses to Questions

**Comments to the Author**

1. Is the manuscript technically sound, and do the data support the conclusions?

Reviewer #1: Yes

Reviewer #2: Partly

2. Has the statistical analysis been performed appropriately and rigorously?

Reviewer #1: Yes

Reviewer #2: Yes

3. Have the authors made all data underlying the findings in their manuscript fully available?

Reviewer #1: Yes

Reviewer #2: No

4. Is the manuscript presented in an intelligible fashion and written in standard English?

Reviewer #1: Yes

Reviewer #2: Yes

Reviewer #1: Using a descriptive phenomenological design, the study aimed to investigate and characterize professional Ghanaian football players' viewpoints on the function of mental toughness in their decision-making. My observations are given below:

(1) The study title is okay.

(2) The abstract needs revision. Although the abstract is informative and well-structured, it is extremely detailed, particularly when it comes to reporting age statistics and extensive themes. Clarity and scholarly appeal would be enhanced by condensing results while highlighting novelty, psychological contribution, and wider implications.

(3) The introduction also needs minor revision. A thorough overview of mental toughness, its conceptual difficulties, and its application to decision-making is given in the introduction. But because it is so long and contains so much detail, it loses focus. The shift from general literature to the specific research gap is slow, despite the fact that the cultural significance of football in Ghana is well-contextualized. Clarity would be increased by more precisely stating the gap, novelty, and particular goals. Conciseness is also weakened when conceptual problems are repeated. A clearer structure would improve readability and more successfully draw attention to the study's contribution.

(4) The method again needs minor amendments. The thorough materials and methods section provides a clear explanation of sampling, data collection, and analysis as well as a compelling philosophical defense of descriptive phenomenology. Trustworthiness tactics and ethical issues are thoroughly explained. However, the theoretical exposition in this section could be simplified to make it easier to read. Reliance on coaches as gatekeepers may introduce bias and should be more critically acknowledged, even though purposive sampling and data saturation are justified. Methodological rigor would be further improved by greater clarity regarding reflexivity procedures and limitations.

(5) The results may be improved. Rich, well-organized, and full of detailed participant quotes, the results section serves as a solid foundation for the thematic analysis. It is admirable how lower- and higher-order themes are integrated to highlight the complex aspects of mental toughness. However, the section lacks focus and clarity due to its excessive length and occasional repetition. Without sacrificing depth, some themes could be condensed into a shorter summary. Additionally, although tables are helpful, there is room for improvement in how they are incorporated into the story. This section would be strengthened by more critical interpretation in addition to descriptive reporting.

(6) The discussion needs minor improvements. The conversation offers a comprehensive, culturally informed explanation of Ghanaian football players' mental toughness and emphasizes its multifaceted nature beyond performance improvement. Although the section is extremely descriptive and shows little critical engagement with contradictions or alternative explanations, the integration of existing literature is praiseworthy. A stronger theoretical contribution would result from a greater focus on how findings expand, improve, or contradict accepted theories. Furthermore, even though they were mentioned, the implications for applied practice could be developed more methodically. Clarity and scholarly impact would be improved by tightening the narrative.

Reviewer #2: Comment 1:

The literature review in the introduction lacks logical coherence and clear conceptual groundwork. Specifically, in lines 30–37, the author discusses the importance and role of "psychological resilience" without first providing a clear definition or conceptual explanation, resulting in insufficient logical rigor. Additionally, the cited literature is predominantly from 2000–2015, lacking recent research findings. The author is advised to:1�Before discussing the significance of psychological resilience, first clearly define its concept; 2�Reorganize the logical sequence of the literature review to better align with the structure of scientific argumentation; 3�Incorporate relevant research findings from the past 5 years, particularly literature linking psychological resilience to decision-making processes; 4�Further demonstrate why "decision-making" is a key entry point for understanding psychological resilience.

Comment 2:

The research methodology section is relatively limited, relying solely on interviews for data collection and lacking supplementary methods that could enhance the validity and generalizability of the findings. A single method may result in insufficient explanatory power for conclusions and make it difficult to capture more direct relationships between psychological resilience and decision-making behaviors. The authors are advised to consider the following in future research designs:1� Incorporate quantitative scales, such as psychological resilience assessment tools (e.g., Mental Toughness Questionnaire 48, etc.); 2�Combine situational or behavioral tasks (e.g., simulated decision-making scenarios, competition strategy selection tasks) to obtain more objective indicators of decision-making performance; 3�Enhance the reliability and external validity of results through mixed-methods design. 4�It is recommended to provide specific evidence supporting data saturation.

Comment 3:

All participants in the current study were drawn from a sample of professional football players in Ghana, resulting in an overly concentrated sample source that may, to some extent, limit the external validity and generalizability of the conclusions. Mental resilience and decision-making processes are significantly influenced by cultural, socioeconomic backgrounds, and competitive systems. Relying solely on samples from a single country or cultural group may make it difficult to support conclusions with broad applicability.However, Ghanaian athletes as research subjects also hold certain value. For instance, their competitive cultural background, resource-constrained environment, and social pressure characteristics can provide an important supplement to mental resilience research that differs from samples in Europe and North America.Therefore, the authors are advised to:

1� Clearly articulate the rationale and limitations of sample selection in the paper, particularly the potential influence of Ghanaian athletes' cultural and environmental characteristics on mental resilience and decision-making;

2�. Consider including athletes from other countries or diverse cultural backgrounds in future research to enhance external validity;

3�. If continuing to use only Ghanaian samples, strengthen the theoretical explanation of "context specificity" to avoid overgeneralizing conclusions.

Comment 4:This paper lacks an in-depth exploration of the mechanisms through which psychological resilience influences decision-making. Current discussions on their relationship remain at the phenomenological level, lacking theoretical explanations and mechanistic elaboration, which limits the theoretical contribution of the study. The authors are advised to further investigate the potential psychological processes by which resilience may affect decision-making behaviors (e.g., stress regulation, attentional control, self-efficacy, cognitive flexibility, etc.) and introduce relevant theoretical frameworks (e.g., dual-process theory, self-regulation theory, etc.) for argumentation. Even if empirical validation is not immediately feasible, proposing a conceptual model or mechanistic hypothesis could enhance the theoretical depth and academic value of the paper.

This manuscript presents a topic with innovation and practical significance, particularly in integrating cultural context differences with psychological resilience—decision-making mechanisms. However, the current draft leans more toward doctoral dissertation outcomes and requires substantial improvements in theoretical construction, methodological transparency, logical coherence of results and discussion, as well as language refinement. A major revision is recommended before reconsideration.

**Do you want your identity to be public for this peer review?** For information about this choice, including consent withdrawal, please see our Privacy Policy

Reviewer #1: **Yes:**  Gyanesh Kumar Tiwari

Reviewer #2: No

---

## [Author Response · Author response to Decision Letter 1]

2 Jan 2026

We are grateful for the opportunity to revise and resubmit our manuscript following the helpful feedback from reviewers. We have carefully addressed all reviewer comments and editorial requests through substantial manuscript revisions and have prepared a detailed point-by-point Response to Reviewers document.

Summary of Revisions:

Reviewer 1: We addressed all comments through comprehensive revisions including: (1) restructuring the abstract to emphasize novelty and theoretical contribution while reducing word count; (2) streamlining the introduction, updating literature with recent references (2020-2024), and strengthening the research gap and novelty statements; (3) simplifying philosophical methodology sections, enhancing reflexivity documentation, and strengthening data saturation evidence; (4) improving results presentation with better interpretive depth while maintaining phenomenological integrity; and (5) substantially revising the discussion to add mechanistic explanations, theoretical frameworks, critical engagement with existing theories, and condensed applied implications.

Reviewer 2: We clarified the conceptual distinction between mental toughness and psychological resilience throughout our responses, and addressed substantive concerns regarding: (1) introduction structure and literature currency; (2) phenomenological methodology appropriateness and quality criteria; (3) cultural specificity as methodological strength rather than limitation, with comprehensive analysis of context-specific versus potentially universal findings; and (4) insufficient mechanistic elaboration, which we addressed through systematic integration of psychological mechanisms (executive control, emotional regulation, attentional control, cognitive flexibility) and theoretical frameworks (self-regulation theory, executive function theory, emotion regulation theory) throughout the revised discussion.

Editor's Data Availability Request: We have provided non-author institutional contact information (Dr. Ernest Adankwah, KNUST), clarified ethical restrictions preventing public data sharing, specified that anonymized data are available upon reasonable request, and detailed long-term data storage and retention procedures in accordance with ethics committee requirements.

A comprehensive, point-by-point Response to Reviewers document accompanies this resubmission, detailing how each comment was addressed with specific references to revised manuscript sections. We have also included a tracked-changes version of the manuscript to facilitate review of all modifications.

We believe these revisions have substantially strengthened the manuscript's theoretical contribution, methodological rigor, and scholarly quality. We remain grateful for the reviewers' constructive feedback and the editorial team's guidance throughout this process

---

## [Decision Letter · Decision Letter 1]

28 Jan 2026

A Phenomenological Exploration of Mental Toughness in Decision-making: Perceptions from Professional Ghanaian Footballers

PONE-D-25-33728R1

Dear Dr. Asamoah,

We’re pleased to inform you that your manuscript has been judged scientifically suitable for publication and will be formally accepted for publication once it meets all outstanding technical requirements.

Kind regards,

Wanli Zang, Ph.D.

Guest Editor

PLOS One

Additional Editor Comments (optional):

Reviewers' comments:

Reviewer's Responses to Questions

**Comments to the Author**

Reviewer #1: All comments have been addressed

2. Is the manuscript technically sound, and do the data support the conclusions?

Reviewer #1: Yes

3. Has the statistical analysis been performed appropriately and rigorously?

Reviewer #1: Yes

4. Have the authors made all data underlying the findings in their manuscript fully available?

Reviewer #1: Yes

5. Is the manuscript presented in an intelligible fashion and written in standard English?

Reviewer #1: Yes

Reviewer #1: The revised manuscript makes a significant contribution. The study is noteworthy because it offers the first phenomenological proof of how professional Ghanaian football players' decision-making is influenced by mental toughness. It expands mental toughness theory beyond performance enhancement to include career planning, leadership, life management, and culturally embedded decisions by emphasizing players lived experiences. The results provide administrators, sport psychologists, and coaches with context-sensitive insights to create psychological support systems and talent development programs that are culturally sensitive in African professional football contexts and policy frameworks.

**Do you want your identity to be public for this peer review?** For information about this choice, including consent withdrawal, please see our Privacy Policy

Reviewer #1: **Yes:**  Gyanesh Kumar Tiwari

---

## [Editor Report · Acceptance letter]

PONE-D-25-33728R1

PLOS One

Dear Dr. Asamoah,

I'm pleased to inform you that your manuscript has been deemed suitable for publication in PLOS One. Congratulations! Your manuscript is now being handed over to our production team.

Kind regards,

on behalf of

Dr. Wanli Zang

Guest Editor

PLOS One